# The Gut Microbiome from a Biomarker to a Novel Therapeutic Strategy for Immunotherapy Response in Patients with Lung Cancer

Sreya Duttagupta [1], Taiki Hakozaki [1,2], Bertrand Routy [1,3,*] and Meriem Messaoudene [1,*]

1 University of Montreal Research Centre (CRCHUM), Montreal, QC H2X 0A9, Canada; sreya.duttagupta@umontreal.ca (S.D.); t-hakozaki@akane.waseda.jp (T.H.)

2 Graduate School of Advanced Science and Engineering, Faculty of Science and Engineering, Waseda University, Tokyo 169-8050, Japan

3 Hematology-Oncology Division, Department of Medicine, University of Montreal Healthcare Centre, Montreal, QC H2X 3E4, Canada

* Correspondence: bertrand.routy@umontreal.ca (B.R.); meriem.messaoudene@umontreal.ca (M.M.)

**Abstract:** The gastrointestinal microbiome has been shown to play a key role in determining the responses to cancer immunotherapy, including immune checkpoint inhibitor (ICI) therapy and CAR-T. In patients with non-small cell lung cancer (NSCLC), increasing evidence suggests that a microbiome composition signature is associated with clinical response to ICIs as well as with the development of immune-related adverse events. In support of this, antibiotic (ATB)-related dysbiosis has been consistently linked with the deleterious impact of ICI response, shortening the overall survival (OS) among patients on ATBs prior to ICI initiation. In parallel, several preclinical experiments have unravelled various strategies using probiotics, prebiotics, diet, and fecal microbiota transplantation as new therapeutic tools to beneficially shift the microbiome and enhance ICI efficacy. These approaches are currently being evaluated in clinical trials and have achieved encouraging preliminary results. In this article, we reviewed the recent studies on the gut microbiome as a potential biomarker and an adjuvant therapy to ICIs in NSCLC patients.

**Keywords:** non-small cell lung cancer; immune checkpoint inhibitors; gut microbiome; antibiotics; probiotics; prebiotics; polyphenols; fecal microbiota transplantation; diet

## 1. Introduction

Among all the molecular pathways involved in the development of a neoplasm, the inherent ability of cancer cells to escape immune response is the most important factor in their growth and has simultaneously revolutionized cancer treatment [1]. The advent of immune checkpoint inhibitors (ICIs) with anti-cytotoxic T-lymphocyte-associated antigen 4 (CTLA-4) and anti-programmed cell death protein 1 (PD-1)/programmed cell death ligand 1 (PD-L1) has provided unprecedented efficacy gains in numerous cancers including non-small cell lung cancer (NSCLC). For patients with high PD-L1 when anti-PD-1 is used as a first-line monotherapy, median overall survival (OS) has nearly doubled (26.3 months vs. 13.4 months), and the 5-year survival has also increased (31.9% vs. 16.3%) in comparison to conventional cytotoxic chemotherapy. Even for the population with low or negative PD-L1, the combination of anti-PD-L1 and chemotherapy yielded a median OS of over 20 months and a 5-year survival of 20% [2,3]. Now, ICI therapy is established as a standard of care for NSCLC irrespective of histological subtype [2,4–6]. ICIs have also been observed to load a long-term memory T-cell response that decreases the likelihood of recurrence of the disease [7]. Therefore, ICIs have been shown to be highly effective in both adjuvant and neoadjuvant settings [8]. Despite the widespread success of ICIs, a majority of the patients develop primary or secondary resistance and immune-related adverse events (irAEs),

representing a major hurdle in cancer immunotherapy [9]. Thus, there is an unmet clinical need to discover predictive biomarkers and novel strategies to increase ICI response.

The gastrointestinal tract harbors trillions of microorganisms [10]. In humans, these microbes alone amount to more than ten times the number of cells in the body [10]. The gut microbiome, representing all the intestinal microorganisms along with their genes, encoded proteins, and cofactors, has a major impact on the physiological processes in their host's body [11]. This unique ecosystem majorly affects the host immune system's homeostasis, leading to inflammatory response beyond the gastrointestinal tract [12]. Indeed, a lower bacterial diversity has been directly linked to several chronic diseases such as inflammatory bowel disease (IBD), celiac disease, obesity, diabetes, and cardiovascular disease [12–15].

From the oncology perspective, it has been recognized that certain bacteria, such as *Helicobacter pylori*, have a direct impact on the host. They bind to gastric epithelial cells with the help of adhesin HopQ, engaging a carcinoembryogenic-related cell adhesion molecule to translocate its virulent factor, cytotoxin-associated gene A (CagA), into the host cell. Then, CagA binds to cellular proteins such as SHP-2 in the cytoplasm, which then activates downstream signaling pathways such as ERK/MAPK. Altogether, this process increases the expression of apoptotic proteins B-cell lymphoma 2 (Bcl-2) and B-cell lymphoma-extra large (Bcl-XL) [16]. This promotes cell proliferation and inhibits apoptosis, leading to gastric cancer and mucosa-associated lymphoid tissue (MALT) lymphoma [16]. Approximately 20% of human malignancies are attributed to specific individual microbes including *H. pylori*, hepatitis B and C, and human papillomavirus, causing gastric, liver, and cervical and uterine cancers [17,18].

Importantly, evidence also suggests that not only one bacterium but an altered microbiome ecosystem, referred to as dysbiosis, may alter the global production of metabolites. Skewed metabolite composition, including the downregulation of butyrate production, can also be directly linked to DNA damage and a disrupted cell cycle, promoting local carcinogenesis and colon cancer development [19]. The discovery of the potential role of the gut microbiome in regulating the host's immune defenses has also shown an important impact on response to cancer therapeutics. With the help of next-generation DNA sequencing techniques, advancements in bioinformatics, and murine germ-free (GF) experiments, the composition of the gut microbiome has been extensively studied over the past eight years, and an avalanche of papers has contributed to demonstrating the link between microbiome and immunotherapy, including CAR T-cells and ICIs [20,21].

Several microbiome profiling studies have established the correlation between gut commensal microbes such as *Akkermansia muciniphila*, *Faecalibacterium prausnitzii*, and *Bifidobacterium* spp. and the response to ICIs, in different malignancies [22–25]. A recent meta-analysis integrating more than 800 patients defined an oncomicrobiome signature for ICI response or resistance. Indeed, several bacteria, including *Enterocloster* and *Hungatella*, have consistently been found to be increased in non-responder patients. Nevertheless, the mechanism explaining how the microbiome affects the therapeutic actions of ICI therapy still remains largely unknown. One of the first pioneering studies performed in 2015 demonstrated that the efficacy of the anti-CTLA-4 antibody was higher in specific pathogen-free (SPF) mice compared to GF or antibiotic (ATB)-depleted mice. In this study, Vetizou et al. showed that recolonization of GF mice with specific commensals such as *Bacteroides fragilis* and *Bacteroides thetaiotaomicron* restored the anti-CTLA-4 activity [26]. The majority of the studies showed that the activities in the microbiome occur through antigen-specific mechanisms, where *Enterococcus hirae* bacteria and the tumor antigens share common epitopes that can affect the anti-tumor immune response. Other groups also demonstrated antigen-independent mechanisms, such as modulating the innate and/or adaptive immune cells, that directly impacted the ICI response [26–28].

The role of the microbiome extends further than just a biomarker for response. Murine supplementation with probiotics such as *A. muciniphila*, *Bifidobacterium* and a consortium of 11 bacterial species, prebiotics such as castalagin, and FMT from renal cell carcinoma (RCC) patients in complete response have all been shown to decrease primary resistance. These

gut modulatory techniques demonstrate the role of the microbiome in ICI treatments and thus open newer therapeutic avenues. These results have subsequently been translated into clinical trials in patients with melanoma [29,30]. Early success from these trials in which patients undergo FMT has propelled the microbiome to be considered the "immune-cancer set point" [21], and, since 2022, one of the "hallmarks of cancer" [31].

In this review, we highlight the recent literature that demonstrates this paradigm shift. We review the literature on how the microbiome predicts response to immunotherapy and how it is negatively altered post-ATB, and we also discuss novel strategies that are currently being investigated in trials to improve microbiome composition.

## 2. Microbiota Profiling of Patients with NSCLC

### 2.1. Gut Microbiome in NSCLC Patients

Extensive gut microbiome profiling through 16S rRNA sequencing and shotgun metagenomics has established that several commensal bacteria in the gut are critically important for ICI response.

The first large trial showed that in 100 patients with NSCLC and renal cell cancer (RCC) undergoing anti-PD-1 therapy, specific bacteria such as *A. muciniphila* and *Alistipes indistinctus* were overrepresented in the feces of patients achieving at least a stable disease [21,22]. Subsequently, this observation was extended to a cohort of 338 NSCLC patients, whereby the presence of *A. muciniphila* in the stool of patients had a prognostic impact on anti-PD-1 therapy independently of gender, age, and PD-L1 expression [32]. *A. muciniphila* was found in 62% of the patients who achieved an objective response and 48% of those who did not. However, it was also noted that the prognostic benefit was retained only when the relative abundance of the bacteria was below the 77th percentile. When the relative abundance increased further in a state considered high Akkermansia (Akkhigh), the OS was shorter. Interestingly, ATB use had been linked to an overabundance *of Akkermansia* above the 77th percentile [32].

To support the role of *A. muciniphila*, FMT from *A. muciniphila*-negative mice translated into anti-PD-1 resistance. Conversely, in these experiments, oral supplementation with *A. muciniphila* could restore anti-PD-1 activity [33].

The relationship between the presence of *A. muciniphila* in the gut and favorable outcomes with ICI therapy among patients with NSCLC has been supported by studies performed by Grenda et al. [34] In this study, the authors performed NGS on the gut microbiome of 47 NSCLC patients undergoing first- or second-line anti-PD-1/anti-PD-L1 therapy. They found a positive correlation with *Akkermansiaceae* in the gut and disease stabilization and response to immunotherapy. They also detected a higher percentage of this family of bacteria in squamous cell carcinoma in comparison to adenocarcinoma.

Furthermore, Liu et al. [35] found *Akkermansia* as one of the dominant genera in patients who tested positive for cytokeratin 19 fragments (CYFRA21-1) [35] This gene is mostly expressed in epithelial-derived cells and thus has a higher expression in lung squamous cell carcinoma compared to lung adenocarcinoma and small cell lung carcinoma.

Newsome and colleagues performed 16S rDNA and RNA sequencing on the fecal samples of 65 NSCLC stage III/IV patients in the United States, both pre- and post-ICI therapy, and categorized the patients as responders and non-responders according to the RECIST criteria [36]. Other than *Akkermansia*, they found the genera *Ruminococcus* and *Faecalibacterium* to be particularly high in responders. Consequently, FMT from responders to gnotobiotic mice decreased tumor growth in comparison to gnotobiotic mice that underwent FMT from non-responders. Similarly, Dora and colleagues studied 62 advanced NSCLC patients who were on anti-PD-L1 therapy. Patients with progression-free survival (PFS) < 6 months had elevated levels of Firmicutes and Actinobacteria phyla, along with a positive correlation with *Streptococcus salivarius*, *Streptococcus vestibularis*, and *Bifidobacterium breve*. Patients with PFS > 6 months, on the other hand, were associated with a higher abundance of *Alistipes spp*. and *Barnesiella visceriola* [37]. Furthermore, another study on metastatic colorectal cancer patients and NSCLC patients treated by cetuximab + avelumab

showed that fecal samples of long-term responders had particularly higher populations of two butyrate-producing bacteria, *Agathobacter M104/1* and *Blautia SR1/5* [38]. This is especially significant as it opens up the possibility of finding a butyrate-mediated immune response mounted by the host on the malignant cells.

In Japan, three separate analyses of the gut microbiome of advanced NSCLC patients showed a distinctive gut microbiome composition in responders [39–41]. First, Katayama et al. retrospectively examined a small cohort of 17 advanced NSCLC patients receiving ICIs for more than 3 months. Responders, who had prolonged time to treatment failure (TTF) had significantly higher levels of *Lactobacillus*, *Clostridium*, and *Syntrophococcus*, while non-responders' guts were found to be enriched with *Bilophila*, *Sutterella*, and *Parabacteroides* [39]. The second study prospectively studied fecal samples of 70 NSCLC patients, 16 of whom had received ATBs 1 month prior to ICI therapy. Both ATB-treated patients and ATB-free patients who were non-responders were found to have a lower *α*-diversity at baseline. Patients who had an OS of more than 12 months (responders) had a gut microbiome enriched with *Ruminococcaceae UCG 13* and *Clostridiales* order, which were particularly underrepresented in ATB patients. They also noted that *Lactobacillaceae* and *Raoultella* were enriched in patients who experienced no significant irAE [40]. In the third study, the authors analyzed the gut microbiome of 28 NSCLC patients and reported similar results. *α*-diversity was significantly lower in non-responders' guts in comparison to that of responders; however, the *β*-diversity was significantly lower in responders [41]. Similar to Martini et al. [38], they also found *Blautia* to be one of the main constituents of the responders' guts. Bacterial species such as *A. muciniphila*, *Allistipes* spp., *Lactobacillus*, *Blautia*, and *Bifidobacterium* spp. have been repeatedly associated with positive response, whereas *Fusobacterium* spp. and *Parabacteroides diastonis* are linked to ICI resistance. These key findings indicate that defining the microbiota of responders in contrast to non-responders is critical in evaluating the potential of the gut microbiome as a biomarker for immunotherapy. The taxonomic composition of responding vs. non-responding patients to ICI therapy is summarized in Table 1.

**Table 1.** Microbiome profiling of responding vs. non-responding NSCLC patients treated with ICIs.

| References | Sample | Type of Study | ICI | Stage of NSCLC | N | Technique | Responders | Non Responders | Notes | Country |
|---|---|---|---|---|---|---|---|---|---|---|
| (Routy et al., 2018) [22] | Feces | Retrospective | Anti-PD-1 | All stages | 153 | WGS | Higher: *Akkermansia muciniphila*, *Ruminococcus* spp., *Allistepes* spp. and *Eubacterium* spp. | Higher: *Parabacteroides distasonis*, *Bacteriodes nordii* | ATB uptake negatively impacts OS, but proton pump inhibitor did not. | France |
| (Derosa et al., 2022) [32,33] | Feces | Prospective | Anti-PD-1 | All stages | 338 | WGS | Higher: *A. muciniphila*, *Eubacterium hallii*, and *Bifidobacterium adolescentis* | Higher: *Clostridium innoccuum* | Stools with *Akkermansia* above the 77th percentile is deleterious. | France, Canada |
| (Newsome at al., 2022) [36] | Feces | Prospective | Anti-PD-1/PD-L1 or anti-PD-L1 and anti-CTLA-4 combination | Advanced | 65 | 16S rRNA (V1-V3) | Higher: *Ruminococcus*, *Akkermansia*, *Blautia*, and *Faecalibacterium* | NA | RNAseq on fecal RNA (N = 10) showed different bacterial transcriptomes within responders and non-responders, such as carbon fixation pathway enriched in prokaryotes in responders while non-responders were enriched in phosphotransferase system. | United States |
| (Martini et al., 2022) [38] | Feces | Prospective | Anti-PD-L1 | Advanced | 10 | 16S rRNA (V4) | Higher: *Agathobacter M104/1* and *Blautia SR1/5* | NA | All patients received ICI as cetuximab + avelumab combination. | Italy |
| (Katayama et al., 2019) [39] | Feces | Retrospective | Anti-PD-1 | Advanced | 17 | 16S rRNA (V1-V2) | Higher: *Lactobacillus*, *Clostridium*, and *Syntrophococcus* | Higher: *Sutterella*, *Bilophila* and *Parabacteroides* | Patients with higher abundance of *Lactobacillus* and *Clostridium* also had longer treatment to TTF. | Japan |
| (Hakozaki et al., 2020) [40] | Feces | Prospective | Anti-PD-1/PD-L1 | Advanced | 70 | 16S rRNA (V3-V4) | Higher: *Agathobacter* and *Ruminococcaceae UCG 13* | Higher: *Eggerthellaceae* and *Barnesiella* | ATB use was associated with lower α-diversity. *Lactobacillaceae* and *Raoultella* were enriched in patients with no significant irAE. | Japan |
| (Shoji et al., 2021) [41] | Feces and Saliva | Prospective | Anti-PD-1/PD-L1 | Stage II/III/IV | 28 | 16S rRNA (V3-V4) | Higher: *Blautia* | Higher: *RF32 unclassified* | Responders had higher α-diversity but lower β-diversity in feces. No significant signal was found from saliva. | Japan |

| References | Sample | Type of Study | ICI | Stage of NSCLC | N | Technique | Responders | Non Responders | Notes | Country |
|---|---|---|---|---|---|---|---|---|---|---|
| (Ouaknine et al., 2018) [42] | Blood | Prospective | Anti-PD-1 | Advanced | 35 | 16S rRNA (V3-V4) | Higher: *Peptostreptococcaceae, Lewinella, Paludibaculum,* and *Holophagae* | Higher: *Gemmatimonadaceae* | Presence of *Gemmatimonadaceae* at baseline was associated with worse PFS and OS. | France |
| (Jin et al., 2019) [43] | Feces | Prospective | Anti-PD-1 | Advanced | 37 | 16S rRNA (V3-V4) | Higher: *Alistipes putredinis, Bifidobacterium longum,* and *Prevotella copri* | Higher: *Ruminococcus_unclassified* | Responders had higher α-diversity. High α-diversity was associated with enhanced memory T cell and NK cell signatures. | China |
| (Song et al., 2020) [44] | Feces | Prospective | Anti-PD-1 | Advanced | 63 | WGS | Higher: *Parabacteroides* and *Methanobrevibacter* | Higher: *Veillonella, Selenomonadales,* and *Negativicutes* | Responders had higher β-diversity. Differences in KEGG functional group and metabolic potential of methanol and methane were also noted. | China |
| (He et al., 2021) [45] | Feces | Prospective | Anti-PD-1 | Advanced | 16 | 16S rRNA (V3-V4) | Higher: *Escherichia, Shigella, Akkermansia,* and *Olsenella* | Higher: *Anaeroglobus* | *Escherichia-Shigella* was positively correlated with IL-12, IFN-γ, and basophils in plasma. *Akkermansia* was positively correlated with monocytes. | China |
| (Zhang et al., 2021) [46] | Feces and Saliva | Prospective | Anti-PD-1 | Advanced | 75 | 16S rRNA (V3-V4) | Higher (in feces): *Desulfovibrio,* Actinomycetales, *Bifidobacterium,* Odoribacteraceae, *Anaerostipes,* Rikenellaceae, *Faecalibacterium,* and *Alistipes* | Higher (in feces): *Fusobacterales, Fusobacteriia, Fusobacterium, Fusobacteria,* and *Fusobacteriaceae* | Responders had higher α-diversity in feces. The abundance of *Streptococcus* in saliva was associated with higher CD8+ T cell density. α-diversity between feces and saliva microbiota was uncorrelated. | China |
| (Masuhiro et al., 2022) [47] | BAL | Prospective | PD-1 | Advanced | 12 | 16S rRNA (V3-V4) | Higher: *Bacteriodetes* | Higher: Proteobacteria | Responders had higher α-diversity and CXCL9 levels in BAL. | Japan |

## 2.2. Lung Microbiome in Lung Cancer Patients

Apart from the gut microbiome, researchers have also been increasingly interested in studying the lung microbiome in relation to lung cancer. The lower airway houses important bacterial populations that are distinctive from the oral, skin, vaginal, and gut microbiomes. They are believed to have migrated from the upper airways and gastrointestinal tract and have also been found to be involved in mucosal immunity and maintaining immune homeostasis [48]. Bronchoalveolar lavage (BAL) or lung tissue samples are collected from patients and analyzed using NGS to study the taxonomic composition of the lung microbiome. Research in this sector is still new, with little literature available, involving single-centered small cohorts of clinical trials. Lee et al. concluded, through a study of 20 lung cancer and 8 benign lung tumor patients, that the BAL fluids from lung cancer patients had a significantly higher abundance of the genera *Veillonella* and *Megasphaera*; thus, they can pose as a good predictor of malignancy in the lung [49]. Furthermore, a study also showed that the microbiota profile shifts in the lungs from healthy non-cancerous to cancerous patients, with a gradual decrease in abundance of *Staphylococcus* and *Dialister* and a gradually increasing trend of the genera *Streptococcus* and *Neisseria* [50]. It has also been reported that the lower airway is enriched with *Streptococcus* and *Veillonella*, which account for the upregulation of ERK and PI3K signaling pathways, the latter being significantly involved in cell proliferation and tumor progression [51]. The authors also reported in 2021, from a study consisting of 148 human subjects, that patients with stage IIIB-IV lung cancer had a higher representation of oral commensals such as *Prevotella*, *Streptococcus*, and *Veillonella* in their lower airway microbiota. At the transcriptomic level, these lung microbiomes also had an upregulation of IL-17, PI3K, MAPK, and ERK, with *Veillonella parvula* being found to be the most closely associated operational taxonomic unit (OTU). To test the causal relationship, the authors induced airway dysbiosis using *V. parvula* in KP mice. They found that the dysbiosis caused an upregulation of PI3K/Akt, ERK/MAPK, IL-17A, IL-6/IL-8, and inflammasome pathways and led to the recruitment of Th17 cells and neutrophils [52]. In squamous cell carcinoma cases, Greathouse and colleagues noted that patients with tumors harboring the epithelial-function-impairing TP53 mutations have a unique bacterial composition rich with certain taxa, including *Acidovorax*, the genus which is typically enriched in smokers [53].

It has been shown in a small trial consisting of NSCLC patients who were on nivolumab monotherapy that 6 out 12 had stable disease or partial response and were categorized as responders while the other 6 had progressive disease and were categorized as non-responders. 16S rRNA gene sequencing of the BAL fluid revealed that at the phylum level, Proteobacteria were significantly reduced and Bacteroidetes increased in responders, in comparison to non-responders [47]. Among the two groups, the $\alpha$-diversity was lower in the non-responders, whereas no such significant difference was observed in the $\beta$-diversity [47]. Similar results regarding $\alpha$-diversity were also obtained by Boesch et al. performing 16S rRNA of the lung tissues of 38 stage III/IV NSCLC patients who received anti-PD-1/PD-L1 treatments [54]. Although early evidence is promising, the role of lung microbiome remains unknown for ICI response; more importantly, its association with the gut microbiome needs to be explored.

## 2.3. Impact of Concurrent Medications on ICI Responses in Cancer

The deleterious impact of ATBs on gut microbiome composition, especially after a course of penicillin, tetracycline, fluoroquinolones, and macrolides, has been previously characterized [55]. They not only act against the pathogenic bacteria but also affect the growth of commensals, inducing the growth and colonization of opportunistic bacteria such as *Clostridium difficile*. In addition, ATB overuse also increases the risk of spreading ATB resistance through horizontal gene transfer [56]. Given the intricate relationship between gut microbiota and immunotherapy efficacy, the detrimental effect of ATBs has the potential to be linked to the latter. In this sense, previous works have already established the role of gut microbes in the efficacy of ICI treatment. The first demonstration that administration of

ATBs prior to initiation of ICI treatment was found to have shorter OS and PFS in NSCLC patients [7,21,57] was published in two large cohorts of 140 and 239 advanced NSCLC patients treated with anti-PD-1/PD-L1 and/or anti-CTLA-4. The authors found drastic differences in the median OS and PFS among patients treated with ATBs (ATB (+)) or not (ATB (-)). Among the first cohort of 140 patients, the median OS was 8.3 months in the ATB (+) group compared to 15.3 months in the ATB-group ($p$ = 0.001). In the second cohort of 239 patients, the median OS was 7.9 vs. 24.6 months, and PFS was 1.9 vs. 3.8 months in the ATB (+) vs. ATB (-) group, respectively ($p \leq 0.01$) [22,58]. Subsequently, several investigations including a large-scale comparative study by Cortellini and colleagues found that ATB exposure significantly reduced the median PFS and OS in NSCLC patients with high PD-L1 expression undergoing pembrolizumab monotherapy [59]. In comparison, no differences were found in ATB-treated patients undergoing cytotoxic chemotherapy.

In addition, a retrospective analysis reported that 176 out of a total of 522 patients of locally advanced NSCLC with prior ATB treatment had shorter OS and PFS compared to the ATB (-) group [60]. This study also highlights the harmful impact of certain classes of ATBs such as β-lactamase inhibitors and quinolones. Interestingly, the shortest median OS was observed in the β-lactamase inhibitor group, followed by fluoroquinolones and then cephalosporins. These ATBs had been previously found to be associated with lower clinical benefits of ICI therapy in both RCC and NSCLC cohorts as well [58]. In this study, it was also noted that the time of ATB exposure also has an impact on ICI efficacy.

The patients who received ATBs 30 days prior to ICIs had worse clinical outcomes in comparison to those who received them 60 days prior to ICIs [58]. In a meta-analysis of 38 studies that included 11,595 patients, it was ascertained that patients taking ATBs, in particular within a month prior to ICI initiation, had worse statistics for mortality [21]. This study confirms that ATBs play a certain role in blocking or providing resistance to the therapeutic effect of ICIs. However, it is yet to be determined whether their effect towards ICI therapy in combination with platinum-based chemotherapy, which is also a widely used standard immunotherapy modality in patients with NSCLC, is detrimental or not.

This cumulative evidence clearly indicates how ATBs, depending on their type, time of administration, and host genetic factors such as tumor PD-L1 expression, negatively impact the overall response to ICI therapy. Although it is not yet known how ATBs have such a pivotal role in immune response, in a recent study Fidelle et al. found that ATBs are responsible for downregulating a cell adhesion molecule known as MAdCAM-1 found in the ileum, Peyer's patch, and mesenteric lymph nodes. On ATB cessation, when the gut microbiota was spontaneously being restored, the MAdCAM 1 levels were still low at the mRNA and protein levels. Ileal bacterial cultures from ATB-treated mice demonstrated an overrepresentation of *Enterocloster* spp. inducing an exodus of α4β7+ Th17 and RORγt+ FoxP3+ regulatory (Tr17) CD4+ T cells towards the tumor. Interestingly, *Enterocloster* spp. was found to be overrepresented in the fecal microbiota of patients and mice who resisted anti-PD-1 therapy. Further, mass spectrometric metabolomics revealed changes for certain bile acids which might be the missing link between *Enterocloster* spp. overpopulation and reduced MAdCAM-1 level, thus paving the way to one plausible mechanism through which ATBs affect ICI efficacy [61]. Altogether, these results characterize the deviated repertoire of the intestinal ecosystem post ATB administration and its impact on ICI outcomes, leading to changes in oncology practice to judiciously prescribe antibiotics [7].

Strategies to overcome ATB-related dysbiosis are currently being investigated. Besides a judicious ATB stewardship, one hypothetical option could be to delay ICI initiation by 30 days in order to let the microbiome recover. Nevertheless, this has never been tested in a clinical trial, and this delay in ICI therapy might be deleterious, especially in patients with large tumor burden. Alternatively, another approach can be to use probiotics to restore the microbiome post-ATB use. However, some reports suggest that probiotics are not fully capable of doing so. Suez et al. reported that probiotics delayed the restoration of the gut microbiome that was depleted through ATBs. They suggested that soluble factors secreted by *Lactobacillus* might have a contribution to this inhibition. Instead, they

found that autologous FMT was more capable of reversing the microbiome to pre-ATB constitution [62]. This inability of probiotics towards successful gut restoration can also be evidenced by the clinical studies that showed that probiotics administered post-ATB or prior to ICI therapy failed to restore microbiota composition and decreased ICI efficacy [62,63].

In our lab, in an attempt to reduce ATB-related dysbiosis, a colon-targeted ATB adsorbent, DAV132, has been designed. In a randomized trial including healthy volunteers, patients received either two broad spectrum ATBs alone for 5 days or in combination with oral DAV132 for 7 days. DAV132 was shown to protect the microbiome composition during ATB treatment. In addition, FMT using the feces of healthy volunteers treated with ATB + DAV132 in tumor murine models amenable to ICIs was able to preserve ICI cancer efficacy compared to mice that received FMT from the feces of patients who had ATBs alone (Messaoudene et al., 2023 under revision).

Additional concurrent co-medications including proton pump inhibitors (PPIs), the most widely used anti-acid medication, are regularly prescribed to cancer patients. They have a direct impact on microbiome composition as it allows the oral microbiome to pass on to the gastrointestinal tract. Interestingly, PPIs have also been shown to reduce OS and PFS in patients undergoing ICI therapy. PPIs and ATBs have both been individually associated with worse PFS in 212 NSCLC patients and even worse PFS when taken together [64]. In a large-scale meta-analysis comprised of 33 studies with 7383 PPI-treated patients and 8574 PPI-free patients undergoing various ICI therapies, it was confirmed that use of PPIs during ICI therapy was associated with decreased OS and PFS (hazard ratio of 1.31 and 1.30, respectively) [64].. Similar to ATBs, the impact of PPIs on ICI therapy has also been linked to the gut microbiome, as increasing evidence has already demonstrated their potential in modulating the gut microbiome, leading to *Clostridium difficile* and other enteric infections [65].

### 3. Modulation of the Gut Microbiome to Improve the Efficacy of Anti-PD-1/PD-L1

#### 3.1. Fecal Microbiota Transplantation

The consistent reports linking beneficial bacteria to positive ICI response shape the gut microbiome as an excellent biomarker. Moreover, various studies have also considered the potential of manipulating or modifying the gut microbiome to improve the efficacy of ICI therapy [22,32,66]. Among all the gut microbiome modulation strategies, FMT represents the most direct method to shift microbiota composition. It is currently approved and has been demonstrated to be very successful for the treatment of refractory *C. difficile* infection [67]. Factors attributed to the success of FMT in refractory *C. difficile* infection (rCDI) patients include different donor-, recipient-, and procedure-related factors [68]. Donor-related factors such as their diet and microbiota richness are of critical importance. It was seen that patients receiving FMT from donors having procarcinogenic bacteria or polyketide synthase island (pks)+ Escherichia coli in the gut also retained the same in the recipient's gut, while FMT from donors who tested negative for these bacteria helped remove these bacteria from the patient's gut [68]. The second recipient-related factors include genetic and immune predisposition in the recipient. It is crucial that the FMT is performed at the stage where there is no mucosal inflammation, as that might hinder proper bacterial colonization [68]. Procedure-related factors such as timing, route of administration, and dose of FMT constitute the third most important facet contributing to the efficacy of FMT [68]. Preclinical studies have already confirmed the prospects of FMT in improving ICI response in NSCLC, RCC, and melanoma undergoing PD-1 therapy in murine models [23,29,58]. Derosa et al. performed FMT from 5 responders and 10 non-responder RCC patients amenable to ICIs on ATB-administered mice 15 days prior to tumor initiation. It was reported that while mice having FMT from responders also responded to ICIs, mice having FMT from non-responders resisted the treatment. Furthermore, oral supplementation with *A. muciniphila* or *B. salyersiae* to the non-responder mice improved treatment benefits [32]. Furthermore, the three studies have also reported remarkable results when FMT was performed on metastatic melanoma patients undergoing anti-PD-1 therapy [30,69]. In the first

phase of a multicenter trial in Canada, FMT was performed on 20 previously untreated advanced melanoma patients, followed by nivolumab or pembrolizumab. Not only was the FMT found to be safe and well tolerated by patients, with no grade 3 adverse effects reported, but the efficacy was also high, with 65% of them reaching the objective response rate (ORR) including 4 (out of 20 patients) gaining complete response [29].

In the two other trials, fecal samples from patients with profound responses were used. In both the in-human clinical trials, they found the taxa associated with good response to the therapy to be enriched in most of the patients participating in the trial. Gene expression and metabolomic profiles had also shown favorable changes in the responders. All three trials demonstrated that the dissimilarity index in the microbiome between patients and their respective donors were lower in responders in comparison to non-responders. Preliminary success in performing FMT in addition to ICI therapy has led to clinical trials in a wide range of cancers, such as melanoma, renal cell carcinoma, gastrointestinal cancer, and lung cancer [70]. However, data on FMT for NSCLC patients on ICIs are very limited. In this respect, continuing with the remarkable success of phase I FMT from healthy volunteers in capsules (NCT03772899) [29], the Canadian team is now conducting a phase II trial of a cohort of 70 patients having advanced NSCLC or melanoma, undergoing FMT along with ICI therapy at multiple centers (NCT04951583). The phase II trial has three arms: uveal melanoma, cutaneous melanoma, and NSCLC. Scheduled to be completed by 2024, the trial uses RECIST criteria to measure the primary outcome (ORR) and secondary outcomes (PFS, OS, and incidents of treatment-related adverse effects and laboratory test abnormalities).

These encouraging results have prompted researchers around the world to perform FMT for clinical benefit in different types of cancer. In Israel, two trials, a phase I (NCT04521075) and phase II (NCT05502913) of FMT by capsules, are also currently being conducted on lung cancer patients treated with ICIs. The first one includes a cohort of 40 patients who have either advanced melanoma or NSCLC (NCT04521075). The second, conducted by Soroka University Medical Center, has already reached phase II with a cohort of 80 advanced lung cancer patients. Prior to FMT, patients would receive, in a randomized manner, an ATB or placebo capsule (NCT05502913). This large-scale clinical trial is expected to be completed by 2028. Two more trials in China (NCT05008861) and Spain (NCT04924374) have 20 participants each and are using capsulized FMT on lung cancer patients treated with ICI therapy. Another large-scale clinical trial is being conducted by M.D. Anderson Cancer Center in the USA with 800 participants that includes NSCLC patients (NCT03819296). Their objective is to ascertain if FMT through modulation of the microbiota can help with the adverse side effects of ICI therapy, which in some cases also cause colitis.

Along with FMT, several other new techniques have been developed that attempt to reconstruct the gut microbiome to yield immune homeostasis systematically. In cancer immunotherapy also, several clinical trials are underway to assess the efficacy of other gut modulatory techniques. Microbial ecosystem therapeutics (MET) is one such alternative to FMT in which, unlike the FMT, where the pan-genome of the donor gut is transplanted, a selected, well-defined mixture of live bacterial strains is isolated from the stool and transplanted to patients. University Health Network in Toronto (Canada) started a trial (NCT03686202) in 2018 using MET-4, a consortium of human-derived bacteria, on patients with different kinds of solid tumors who are on ICI therapy [71]. Outcome measures include gene sequencing to check the changes in the relative abundance of species in stool from baseline to after 12 days (short term) and 24 weeks (long term), as well as ORR, PFS, and standard IHC of tumors. The investigators found an increase in the relative abundance of several MET-4 taxa that are associated with ICI response, such as *Enterococcus* and *Bifidobacterium* in the patients. These engraftments were also associated with an overall decrease in primary bile acids in the plasma and the stools [72]. In another approach, oral restorative microbiota therapy (RMT) capsules are being given to 82 patients with advanced NSCLC at the Masonic Cancer Center, the University of Minnesota, who are on durvalumab along with platinum-based chemotherapy. This randomized double-blind

trial is in phase II and will be completed in 2028 after measuring ORR, OS, PFS, quality of life (QoL), and duration of response (DOR) (NCT04105270).

Although very promising, the idea of using FMT or other microbiota reconstruction techniques to produce a positive therapeutic outcome for ICIs is still in its infancy. We are yet to understand how the strain engraftment occurs between a donor and a patient's gut. Because of heterogeneity in cohorts and different kinds of diseases, we have few trials where the patients experiencing a clinical success had higher similarity with their donor microbiome compared to the rest, but in some other trials this finding was not replicated [68]. Based on the outcome of these ongoing trials, it needs to be evaluated whether FMT, MET, or RMT can be considered as an adjuvant therapy to ICI treatments in patients with NSCLC.

### 3.2. Probiotics

Benefiting from metagenomics sequencing, researchers have identified specific taxa in the intestinal microbiota that have a direct causal link with a favorable outcome to ICI therapy. As previously stated, oral administration of *A. muciniphila* in mice models who had undergone FMT from non-responders has been found to decrease the resistance to ICI therapy posed by the non-responding intestinal microbiome [32]. Additionally, probiotics are also associated with the production of anti-inflammatory cytokines that interrupt the process of carcinogenesis, by phagocytizing cancer cells. Various strains of *Bacteroides*, *Bifidobacterium*, and *Faecalibacterium* have been associated with anti-PD-1 or anti-CTLA-4 efficacy [21]. *B. breve* was found to be significantly higher in patients who had a better PFS than those who did not, among a cohort of Chinese NSCLC patients treated with anti-PD-1, suggesting its presence as a potential indicative biomarker for better prognosis [73]. Certain other species and genera, such as *A. muciniphila*, *Clostridiales*, and *Ruminococcaceae*, are involved in reversing immunosuppression by improving the gut barrier and thus preventing leaky gut [74]. Four retrospective analyses were performed in Japan between 2020 and 2022 on NSCLC patients undergoing ICI therapy in which some of the patients were administered probiotics. The first was performed in 2020 on 118 advanced NSCLC patients, out of whom 39 had received probiotic *Clostridium butyricum* therapy (CBT). The probiotic CBT group had a better PFS and OS in comparison to the non-CBT group. Importantly, among them, 46 patients had also received ATBs at some point within 60 days of initiation of ICI therapy. While the ATB group was not found to be directly linked to worse clinical outcomes, in that group, those who received probiotic CBT had improved PFS and OS compared to those who did not [75]. In 2021, Takada and colleagues published a study on 294 patients from three different study centers out of whom a total of 32 patients had received probiotics. Apart from *C. butyricum*, other probiotics administered to some of the patients included *Bifidobacterium* and antibiotic-resistant lactic acid bacteria. While there was no significant difference in the OS, the PFS, ORR, and disease control were statistically better in the probiotic-administered patients [76]. A bifidogenic live bacterial product containing *C. butyricum* was also administered in a randomized phase I trial containing 30 advanced RCC patients who were on nivolumab plus ipilimumab. It was reported that PFS and ORR were better in patients who had the bifidogenic product in comparison to those who did not, suggesting a positive correlation of the product to clinical outcome in RCC [77]. In the next year, Takada et al. published another study analyzing 95 postoperative recurrent NSCLC patients receiving ICIs, among whom few had received the same strains of probiotics [78]. They were unable to report any statistically significant association between probiotic administration and ICI efficacy. In the fourth study published in 2021, it was found that the incidence of irAEs, which were 46.5% in the non-probiotic group, were reduced to 28.3% in the probiotic group. Analyzing patients from the same time frame as the previous two studies (i.e., January 2016 to mid-2018), however, failed to associate probiotic administration with any statistically significant response to ICI therapy [79]. Similarly, Svaton and colleagues retrospectively analyzed NSCLC patients on nivolumab in the Czech Republic to also find no statistically

beneficial outcome of probiotic Lactobacillus administration [80]. It is important to note that, in both the cases [79,80], the patients in the probiotic administration group were not more than 5% of the total study group.

*Lactobacillus* and *Bifidobacterium* are the two main classes of bacteria that have been widely studied and used as probiotic supplements. They have also been studied and have been part of clinical trials to improve the efficacy of immunotherapy in NSCLC. Shanghai 10th People's Hospital was the first to start a clinical trial in 2019 to test if *Bifidobacterium* enhances the efficacy of platinum-based chemotherapy in advanced NSCLC. The study is scheduled to be completed in 2024 and will assess the PFS and ORR as primary objectives and OS and fecal microbiome as the secondary objectives (NCT03642548). In the USA, a live biotherapeutic strain of *Enterococcus gallinarum* was given to patients of several types of advanced cancer, including NSCLC, in combination with pembrolizumab, in a trial consisting of 132 patients (NCT03637803). In the next year, Xiangya Hospital of Central South University started a single-arm phase I trial to assess the safety of *Bifidobacterium* in stage III resectable NSCLC patients on neo-adjuvant platinum-based chemotherapy + nivolumab (NCT04699721). In the same year, Genome & Company started another phase I trial on several types of cancers including NSCLC, to test whether GEN-001 (Lactococcus lactis), a live biotherapeutic product, can be safely tolerated by patients who are also treated with avelumab (NCT04601402). Probiotic *Lactobacillus Bifidobacterium V9* (Kex02) is being tested in a randomized trial by Jiangxi Provincial Cancer Hospital on NSCLC patients to see if it improves the efficacy of platinum-based chemotherapy + Carilizumab (NCT05094167).

### 3.3. Diet Evaluation and Prebiotics

The role of diet and nutritional status has been given a lot of importance in relation to the efficacy of immunotherapy in a wide range of cancers. Gouez et al. have recently established the relationship between severe malnutrition and lower survival rates in a cohort of French patients of advanced NSCLC treated with immunotherapy [81]. Ketogenic diet and a protein-restricted diet have both shown a reduction in PD-1, PD-L1, and CTLA-4 expressions and increased the efficacy of ICI treatments in animal models [82]. The direct link of a ketogenic diet with an increased abundance of *A. muciniphila* in the gut can be attributed as one of the possible mechanisms for this. Fasting is hypothesized as a possibly effective way to hinder tumor growth as it leads to a reduction in glucose in the blood and the formation of ketone bodies instead, which act as the main replacement biofuel for all the vital organs of the body. This leads to autophagy in the cancer cells with a noticeable shift in the tumor microenvironment, with a decrease in the detrimental FOXP3$^+$ regulatory T cells and a subsequently beneficial increase in CD8$^+$ cytotoxic T lymphocytes [83]. Trials are underway to assess the role of fasting-mimicking diets on advanced lung cancer patients (NCT03709147 and NCT03700437).

Apart from fasting, dietary interventions or additional supplementation can also be employed as a way to boost the impact of immunotherapy on a wide range of solid tumors, including NSCLC. Camu-Camu (*Myrciaria dubia*), an Amazonian berry, is rich in polyphenols and is associated with prebiotic effects that render protection against several metabolic diseases. Among its range of several constituent phytochemicals, a chemically defined polyphenol, castalagin, has been particularly found to mediate anti-PD-1 effects and improve the CD8$^+$ T cells/FOXP3$^+$ regulatory T cells ratio. Oral administration of either the whole berry or castalagin alone has induced shifts in the gut microbiome of SPF mice with an overrepresentation of bacteria associated with immunotherapeutic response, i.e., *Ruminococcaceae* and *Alistipes*. FMT from non-responders that developed a gut microbiome aligned to non-anti-PD-L1 efficacy also underwent a shift in the taxonomic composition with increased abundance of *Bifidobacterium* and *Ruminococcaceae* after administration of Camu-Camu [84]. Further studies on the supplementation of oral castalagin to GF mice could not decrease the resistance, further validating the dependency of castalagin on the microbiome.

In a cohort of 128 melanoma patients, 37 of them reported sufficient dietary intake. The median PFS was found to be higher in these patients. Interestingly, per 5 g increase in daily dietary fiber intake, the risk of progression or death reduced by 30% [63]. For NSCLC, however, a nurse-directed dietary intervention, assessed by the Food Frequency Questionnaire (FFQ), noted that saturated fatty acid intake and not high fiber intake was better related to longer PFS [85]. Short-chain fatty acids such as propionate and butyrate, along with butyrate-producing bacteria, have been consistently found in higher abundance in healthy adults, as well as in lung cancer patients who have responded to ICI therapy [86–89]. A study pertaining to discovering metabolomics signatures in NSCLC patients undergoing anti-PD-1/PD-L1 therapy is underway at the Grenoble Alpes University Hospital in France and hopefully will be able to give more insight in the future about specific dietary compounds in the gut microbiome that can be linked to better treatment response [90]. Clinical trials are also underway that consider the potential effect of certain dietary compounds, such as herbs, berries, flaxseeds, omega 3 fortified supplements, curcumin, soy isoflavones, folate, and vitamin B12, on different lung cancer patients. However, to our knowledge, only two trials are underway that assess the efficacy of nutrition on advanced or metastatic NSCLC patients who are on immunotherapy. The first is an Italian randomized trial that aims to assess the efficacy of Oral Impact®, an oral high-caloric and high-protein, immunonutrient-enriched nutritional liquid supplement, on such patients by checking their PFS, DOR, OS, self-perceived QoL, and serum immunological markers, among others (NCT05384873). The second is a Taiwanese randomized phase II trial examining the effects of fermented soybean extract MicrSoy-20 (MS-20) among advanced NSCLC patients treated with pembrolizumab (NCT04909034).

### 3.4. Other Techniques

Recently, an MHC class-I binding epitope was found at the tail length tape measure protein (TMP) of a prophage in the genome of *E. hirae*. It was discovered that *E. hirae 13144* harbored a bacteriophage that activated the TMP-specific H-2Kb-restricted CD8+ T cell, improving the response to anti-PD-1 immunotherapy in mouse tumor models. In mouse and human models, eventually, the administration of the intestinal microbe Enterococcus containing this bacteriophage was directly associated with improved T cell response and OS, respectively. Lung cancer patients with higher levels of Enterococcal prophage in the stool enjoyed long-term benefits of the anti-PD-1 therapy [27]. This cross-reactivity between tumor and microbial antigens is novel and potentially revolutionary. Fermented food, symbiotic food, and the use of such phage therapy are a few of the techniques that have shown great results in other metabolic disorders such as IBD and ulcerative colitis but have not been studied at length in relation to cancer and immunotherapy. The link between the gut microbiome and ICI therapy in a wide range of solid tumors, especially in NSCLC, warrants further research in the field of exploring and manipulating intestinal microbes towards improving anti-cancer immunotherapies.

### 3.5. Discussion and Conclusions

For the greater part of the twentieth century, the standard pharmacotherapy for NSCLC was platinum-based chemotherapy. The efficacy of such a standard treatment was poor, with a minimal long-term survival rate. Immunotherapy is a relatively newer approach to cancer treatment, and although it is very promising, there are various unknown avenues related to it. For instance, the important role of ATBs has been of scientific interest only for the past decade. A majority of the studies and trials conducted on the impact of ATBs on NSCLC patients undergoing ICI treatments suggest a negative correlation between ATB intake and clinical response to cancer therapy. Although some contrasting evidence has also been published implying that ATBs have improved OS and PFS or have not brought any change at all [57,91–93], these reports are fewer in number. It is important to note that the period of ATB medication (Figure 1) with respect to ICIs and the class of ATBs given [93,94], duration of the exposure [95], and route of administration [96], apart from the

type of ICI [97], host PD-L-1 expression [59,98], and stage of cancer (Figure 2), all seem to
have an important role in the overall clinical success. As Cortellini et al. [99] noted, NSCLC
likely experiences a special benefit from chemoimmunotherapy, which possibly exerts a
synergistic effect from both treatment modes. These might cause a bias when establishing
the role of ATBs in ICI therapy. Nevertheless, large-scale studies from different parts of
the world have otherwise established that ATB exposure, through modulation of the gut
microbiome, has a direct impact on ICI efficacy.

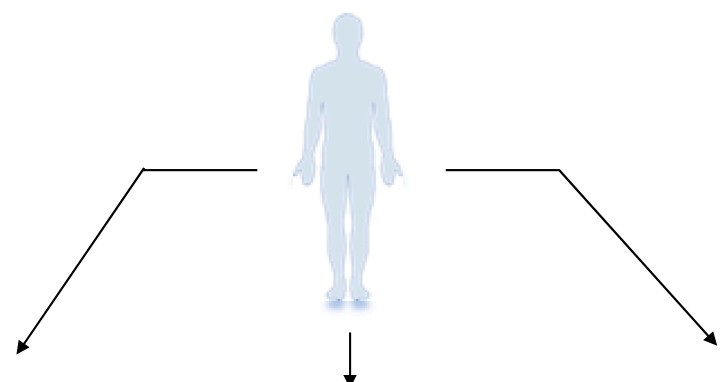

**Host and medication related-Biomarkers**

**Factors influencing antibiotic effectiveness**

✓ Class of antibiotics and routes of administration
✓ Host PD-L1 Expression
✓ Type of ICI
✓ Interaction with other chemotherapeutic drug
✓ Time period and duration of exposure
✓ Stage of the disease

**Tissue/Blood-based biomarkers**

**Tissue**
✓ PD-L1 expression
✓ Tumor mutation burden (TMB)
✓ Tumor-infiltrating lymphocytes (TILs)
✓ Gene expression signatures (T-effector and IFNγ-related gene signature etc.)
✓ Concomitant genomic alterations (EGFR, ALK, STK11/LKB1-comutation etc.)

**Blood**
✓ Blood TMB (bTMB)
✓ Soluble PD-L1 (sPD-L1)
✓ Metabolites
✓ Damage-associated molecular patterns (DAMPs)

**Clinical-based factors**

✓ Neutrophil-to-lymphocyte ratio (NLR), Derived NLR (dNLR)
✓ Lung Immune Prognostic Index (LIPI)
✓ Performance status
✓ Liver metastasis
✓ Presence of massive lesions
✓ Cancer cachexia
✓ Concomitant medications

**Interventions**

✓ Optimization of immunotherapy regimens (ICI only or ICI+Chemo)
✓ Combination with other modalities (αCTLA-4, angiogenesis inhibitor, radiotherapy)
✓ Intervention for cachexia
✓ Deprescribe unnecessary medications

**Figure 1.** Host and medication-related biomarkers that can impact the immune checkpoint blockers'
(ICI) response.

**Microbiome related-biomarkers**

**Response biomarkers**          **Non-Response biomarkers**

<u>Gut microbiome</u>

*Actinomycetales*
*Agathobacter*
*Akkermansia muciniphila*
*Alistipes spp*
*Anaerostipes*
*Bacteriodetes*
*Bifidobacterium longum*
*Blautia*
*Clostridium*
*Desulfovibrio*
*Eubacterium spp*
*Faecalibacterium*
*Holophagae*
*Lactobacillus*
*Lewinella*
*Methanobrevibacter*
*Odoribacteraceae*
*Olsenella*
*Paludibaculum*
*Parabacteroides*
*Peptostreptococcacea*
*Prevotella copri*
*Rikenellaceae*
*Ruminococcus spp*
*Shigella*
*Syntrophococcus*

<u>Metabolites</u>
SCFA, sphingolipids, fatty acyls,
glycerophospholipids in early
NSCLS patients

<u>Lung microbiome</u>
*Streptococcus (Spuctum), Firmicutes*
*(Spuctum), Bacteriodetes (BAL)*

<u>*Saliva microbiome*</u>
*Firmicutes, Lactobacillales,*
*Streptococcus*

<u>Gut microbiome</u>

*Anaeroglobus*
*Bacteriodes nordi*
*Barnesiella*
*Bilophila*
*Clostridium innoccuum*
*Eggerthellaceae*
*Enterocloster*
*Fusobacterium*
*Gemmatimonadaceae*
*Hungatella*
*Negativicutes*
*Parabacteroides distasonis*
*Proteobacteria*
*Selenomonadales*
*Sutterella*
*Veillonella*

<u>Lung microbiome</u>
*Veillonella (in the BAL)*

<u>*Saliva microbiome*</u>
*Proteobacteria*

**Interventions**

✓   Fecal microbiome trasplantation (FMT) (NCT04951583)
✓   Probiotics (*Akkermansia muciniphila*)
✓   Prebiotics (Camu-camu)
✓   Diet support

**Figure 2.** Microbiome-related biomarkers that can impact the immune checkpoint blockers' (ICI) response.

The treatment plan for NSCLC now includes certain blood- and tissue-related biomarkers [100–102] such as tumor mutation burden [103], tumor infiltrating lymphocytes [104], and soluble PD-L1 [105] and other clinical factors such as neutrophil to lymphocyte ra-

tio [106] and lung immune prognostic index [107]. The advent of NGS technology has also helped us understand the influential role of the gut microbiome in relation to disease and therapeutic approaches towards it. The mucin-degrading human commensal *A. muciniphila*, along with some other species of anaerobic commensals such as *Alistipes*, *Lactobacillus*, *Bifidobacterium*, and *Blautia*, are abundantly found in the GM of NSCLC patients who respond well to ICI therapy. Contrastingly, patients who do not respond well to this therapy have been found to have a higher constitution of *Bacteriodes*, *Parabacteriodes*, *Fusobacterium*, and *Clostridium* species in their gut. Microbiota profiling of such patients over the past five years has revealed that most of the studies (8 out of 13) were performed in China and Japan, whereas the other five were performed in Europe and North America. This might result in some bias based on geographic location; hence, a global multicenter cohort of patients is required for obtaining more clarity regarding the gut microbiome of NSCLC patients. The role of short-chain fatty acids, especially butyrate and butyrate-producing gut microbes, is also of great scientific importance, as they have been repeatedly found to be lower in NSCLC patients compared to healthy adults but also higher in responding NSCLC patients compared to non-responding ones [1]. Further mechanistic studies on probiotics are required to fully understand their potential role in immunotherapy in NSCLC patients. Although reports suggest they are not capable of restoring the gut microbiome after ATB-induced gut depletion [56], large-scale studies are still underway to assess their positive effect. It is important to note in this case that the strain of probiotics, and whether they are live or attenuated, also holds a big significance that should be kept under consideration. Studying the effect of probiotics, prebiotics, and other dietary compounds to manipulate the gut into improving the efficacy of immunotherapy and determining their mechanism of action will possibly shed light on some underlying pathway that is linked to the tumor. These studies, along with gut microbiome modulatory techniques such as FMT, MET, phage technology, and probiotics, will help develop a personalized approach to improve the clinical benefit of immunotherapy in cancer.

**Author Contributions:** Writing—original draft preparation, S.D., T.H., B.R. and M.M.; writing—review and editing, S.D., T.H., B.R. and M.M.; supervision, B.R. and M.M. All authors have read and agreed to the published version of the manuscript.

**Funding:** This research received no external funding.

**Conflicts of Interest:** The authors declare no conflict of interest. B.R. received Terry Fox Marathon of Hope clinician-scientist award. M.M. reports salary support from Seerave foundation.

## Abbreviations

| | |
|---|---|
| ALK | Anaplastic lymphoma kinase |
| ATB | Antibiotics |
| BAL | Bronchoalveolar lavage |
| Bcl-2 | B-cell lymphoma 2 |
| Bcl-XL | B-cell lymphoma-extra large |
| CagA | Cytotoxin-associated gene |
| CD8 | Cluster of differentiation 8 |
| CRC | Colorectal cancer |
| CTLA-4 | Cytotoxic T-lymphocyte-associated antigen 4 |
| DNA | Deoxyribonucleic acid |
| DOR | Duration of response |
| EGFR | Epidermal growth factor receptor |
| EMA | European Medicine Agency |
| ERK | Extracellular signal-regulated kinase |
| FFQ | Food Frequency Questionnaire |
| FMT | Fecal Microbiota Therapy |
| FOXP3 | Forkhead box P3 protein |
| GF | Germ-free |

| | |
|---|---|
| GM | Gut microbiome |
| IBD | Inflammatory bowel disease |
| ICIs | Immune checkpoint inhibitors |
| IHC | Immunohistochemistry |
| irAEs | Immune-related adverse effects |
| mAb | Monoclonal antibody |
| MDSC | Myeloid-derived suppressor cells |
| MET | Microbial ecosystem therapeutics |
| MHC | Major histocompatibility complex |
| NGS | Next-generation sequencing |
| NK | Natural killer cells |
| NSCLC | Non-small cell lung cancer |
| ORR | Objective response rate |
| OS | Overall survival |
| PD-1 | Programmed cell death protein 1 |
| PD-L1 | Programmed cell death ligand 1 |
| PFS | Progression-free survival |
| PI3K | Phosphoinositide 3-kinase |
| PPI | Proton pump inhibitor |
| QoL | Quality of life |
| RCC | Renal cell carcinoma |
| rCDI | Refractory Clostridium difficile infection |
| rDNA | Ribosomal deoxyribonucleic acid |
| RECIST | Response evaluation criteria in solid tumors |
| RMT | Oral restorative microbiota therapy |
| rRNA | Ribosomal ribonucleic acid |
| SCLC | Small cell lung cancer |
| SPF | Specific pathogen free |
| TLR | Toll-like receptor |
| TKI | Tyrosine kinase inhibitor |
| TP53 | Tumor protein 53 |
| USFDA | United States Food and Drug Administration |
| VEGF | Vascular endothelial growth factor |

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
