# Peer review of "The Gut Microbiome from a Biomarker to a Novel Therapeutic Strategy for Immunotherapy Response in Patients with Lung Cancer"

_curroncol, doi:10.3390/curroncol30110681_

Round 1

Reviewer 1 Report

Title: The Gut Microbiome From a Biomarker to a Novel Therapeutic Strategy For Immunotherapy Response in Patients With Lung Cancer.

Summary: In this review, authors have discussed most recent updates on role of microbiome in predicting response to immunotherapy. They also shed light on how ATB may impact response to ICI therapy. Authors discussed the recent studies focusing on using gut microbiome as a biomarker and an adjuvant therapy to ICI in NSCLC patients.

Overall I think this review discusses all the clinically relevant aspects of recent updates on role of gut microbiome in response to immunotherapy in NSCLC patients.

Minor comments:

1.      Line 37: “Median overall survival (OS) has increased to approximately 30 months and 5-year survival has increased up to 30% for patients with PD-L1 high when anti-PD-1 is used as first-line monotherapy.” Authors mentioned here that OS and 5-year survival increased to 30 months and 30% respectively. If you are saying increased to; then you have to mention what was it before.

English is fine. Minor spell check is sufficient

Author Response

Dear reviewer,

Thank you for your comments and question, you can find below a reponse to your comment:

Summary: In this review, authors have discussed most recent updates on role of microbiome in predicting response to immunotherapy. They also shed light on how ATB may impact response to ICI therapy. Authors discussed the recent studies focusing on using gut microbiome as a biomarker and an adjuvant therapy to ICI in NSCLC patients. Overall I think this review discusses all the clinically relevant aspects of recent updates on role of gut microbiome in response to immunotherapy in NSCLC patients.

Minor comments:

  1. Line 37: “Median overall survival (OS) has increased to approximately 30 months and 5-year survival has increased up to 30% for patients with PD-L1 high when anti-PD-1 is used as first-line monotherapy.” Authors mentioned here that OS and 5-year survival increased to 30 months and 30% respectively. If you are saying increased to; then you have to mention what was it before.

Our response: We have rephrased this sentence. It now reads as “For patients with PD-L1 high when anti-PD-1 is used as first-line monotherapy, median overall survival (OS) has nearly doubled (26.3 months vs 13.4 months), and the 5-year survival has also increased (31.9% vs 16.3%) in comparison to conventional cytotoxic chemotherapy”.

Reviewer 2 Report

In this review, Duttagupta et al. outline the current evidence for microbiome as a bio-marker for immune checkpoint inhibitor efficacy in NSCLC. They also review current efforts to alter the microbiome through FMT, etc. The authors are to be commended for such a comprehensive review of the literature and insightful comments on future directions.

Author Response

Dear reviewer,

Thank you for your positive comment for this review,

Best regards,

Meriem Messaoudene

Reviewer 3 Report

In this review, Duttagupta S., et al have discussed the role of the gastrointestinal microbiome in cancer immunotherapy, specifically immune checkpoint inhibitor (ICI) therapy and chimeric antigen receptor T-cell (CART-T) therapy, focusing on non-small cell lung cancer (NSCLC) patients. Overall, the review is interesting and gives a detailed review of literature in the field. But this reviewer has a few suggestions to incorporate:

1.       Line 99-103: Rephrase

2.       Expand PFS the first time it is used in the manuscript

3.       Table-1: first row/ under ‘notes’ section/ Change Ab to ATB to maintain consistency

4.       Table-1: Sixth row/ under ‘notes’ section/ Typo: ‘metabolic’

5.       Line 211: Typo: OUT

6.       Section – ‘Lung microbiome…..patients’ seems irrelevant since the theme of the review revolves around gut microbiome

7.       Line 390-394: Rephrase

8.       Line 403: Write a statement about RMT

9.       Line 505-508: Recheck the statement. GF mice doesn’t have the microbiome. Either there is some typo or something is missing.

Overall, English language used in the manuscript is appropriate except at some places which needs rephrasing (see comments)

Author Response

Dear Reviewer,

Thank you for your questions and comments for this review. Please find below, a point-point response to your comments.

Best regards,

Meriem Messaoudene

REVIEWER 3

In this review, Duttagupta S., et al have discussed the role of the gastrointestinal microbiome in cancer immunotherapy, specifically immune checkpoint inhibitor (ICI) therapy and chimeric antigen receptor T-cell (CART-T) therapy, focusing on non-small cell lung cancer (NSCLC) patients. Overall, the review is interesting and gives a detailed review of literature in the field. But this reviewer has a few suggestions to incorporate:
1. Line 99-103: Rephrase

Our response: We have rephrased this paragraph. Now the lines 100-105 reads as “The role of the microbiome extends further than just a biomarker for response. Murine supplementation with probiotics such as A. muciniphila, Bifidobacterium and a consortium of 11 bacterial species, prebiotics such as castalagin and FMT from renal cell carcinoma (RCC) patients in complete response, have all been shown to decrease primary resistance. These gut modulatory techniques demonstrate the role of the microbiome in ICI treatments and thus opens newer therapeutic avenues.”

  1. Expand PFS the first time it is used in the manuscript

Our response: We have expanded the abbreviation PFS in line 151 where it is first used.

  1. Table-1: first row/ under ‘notes’ section/ Change Ab to ATB to maintain consistency

Our response: We have changed “Ab” to “ATB” in Table 1, first row under notes.

  1. Table-1: Sixth row/ under ‘notes’ section/ Typo: ‘metabolic’

Our response: We have changed “meabolic” to “metabolic” in Table 1, sixth row under notes.

  1. Line 211: Typo: OUT

Our response: We have changed “OUT” to “OTU” in line 214.

  1. Section – ‘Lung microbiome…..patients’ seems irrelevant since the theme of the review revolves around gut microbiome

Our response: We understand that our review largely focuses on the gut microbiome, an avenue which currently is being explored in oncology with great interest. However, since our review is particularly discussing on non-small cell lung cancer, we felt it to be important to mention the microbiome of the lung. The readers will be able to appreciate the potential role of the lung microbiome in oncogenesis and its potential intricate interplay between the lung and gut microbiomes in the context of lung cancer development and progression.

  1. Line 390-394: Rephrase

Our response: We have rephrased this paragraph. Now the lines 389-394 reads as “In cancer immunotherapy also, several clinical trials are underway to assess the efficacy of other gut modulatory techniques. Microbial Ecosystem Therapeutics (MET) is one such alternative to FMT in which, unlike the FMT, where the pan-genome of the donor gut is transplanted, a selected, well-defined mixture of live bacterial strains is isolated from the stool, and transplanted to patients.”

  1. Line 403: Write a statement about RMT

Our response: We tried our best to find more details about RMT, however, there is no information provided in the internet regarding its constitution. We also did not find any paper related to this trial, as it is still ongoing (Clinical trial no. NCT04105270).

  1. Line 505-508: Recheck the statement. GF mice doesn’t have the microbiome. Either there is some typo or something is missing.

Our response: We have changed “GF” in that line to “SPF”. We have later added another line (line 509-511), “Further studies on the supplementation of oral castalagin to GF mice could not decrease the resistance, further validating the dependency of castalagin on the microbiome.”